# Automated computed tomography based parasitoid detection in mason bee rearings

**Bart R. Thomson** [1,2], **Steffen Hagenbucher** [3], **Robert Zboray** [4], **Michelle Aimée Oesch** [5], **Robert Aellen** [6], **Henning Richter** [7] *

**1** Department of Neurosurgery, Clinical Neuroscience Center, Universitätsspital and University of Zurich, Zurich, Switzerland, **2** Division of Internal Medicine, Universitätsspital and University of Zurich, Zurich, Switzerland, **3** Institut für Agrarökologie, Agroecology, Science, Aarau, Switzerland, **4** Center for X-ray Analytics, EMPA, Dübendorf, Switzerland, **5** Vetcom, University of Zurich, Zurich, Switzerland, **6** Wildbiene + Partner AG, Zurich, Switzerland, **7** Diagnostic Imaging Research Unit, University of Zurich, Zurich, Switzerland

☙ These authors contributed equally to this work.

* henning.richter@uzh.ch

**Data Availability Statement:** The data that has been used in this publication is publicly accessible under the following DOI: https://doi.org/10.5281/zenodo.6977306.

## Abstract

In recent years, insect husbandry has seen an increased interest in order to supply in the production of raw materials, food, or as biological/environmental control. Unfortunately, large insect rearings are susceptible to pathogens, pests and parasitoids which can spread rapidly due to the confined nature of a rearing system. Thus, it is of interest to monitor the spread of such manifestations and the overall population size quickly and efficiently. Medical imaging techniques could be used for this purpose, as large volumes can be scanned non-invasively. Due to its 3D acquisition nature, computed tomography seems to be the most suitable for this task. This study presents an automated, computed tomography-based, counting method for bee rearings that performs comparable to identifying all Osmia cornuta cocoons manually. The proposed methodology achieves this in an average of 10 seconds per sample, compared to 90 minutes per sample for the manual count over a total of 12 samples collected around lake Zurich in 2020. Such an automated bee population evaluation tool is efficient and valuable in combating environmental influences on bee, and potentially other insect, rearings.

## Introduction

Animal husbandry has been an important part of human development as we heavily rely on animals for food and raw materials [1]. However, human effort was mostly focused on vertebrates, while insects (or other arthropods) were rarely reared and domesticated, with a few exceptions such as the silk moth or the honeybee. In the last decades, there has been an increased interest in utilizing insects to satisfy human needs [2]. Especially in agriculture insects can play an important part as alternative more sustainable plant protection method (e.g., parasitoids and lacewings), pollinators (e.g., bumblebees and solitary bees) or as food and feed (e.g., black soldier flies) [2]. Mass rearing of insects for biological control, production of raw materials or as a food source is now common all over the globe and there are no signs that

**Funding:** The author(s) received no specific funding for this work.

**Competing interests:** The authors do not declare any competing interests.

this trend is slowing down [2]. Already the industry of apiculture has been shown to deliver at least 22 billion € to the European agricultural sector through pollination [3]. However, the successful use of these insects, especially when applied to the agricultural sector, depends on a high quality of the produced animals as they need to healthy and capable of completing their task in the field [2],

As with other animals, a large rearing of insects is susceptible to pathogens and pests [4]. Due to the confined nature of a rearing system, these can spread quickly and severely reduce yield or even destroy such a rearing. A unique threat to insect rearings are parasitoids. A parasitoid lives in close association with its host (either inside or on it) and, in contrast to a parasite, sooner or later kills its host [5]. Most parasitoids are Hymenoptera (so called parasitic wasps), but they are also found in other groups like the Dipterans. Normally a parasitoid lays its eggs inside a host, where the larvae emerge and consume the host [6]. Due to this lifestyle and their small size, most parasitoids are hard to detect and could remain undetected in a commercial rearing, until immense damage has been done [7]. Female parasitic wasps can lay several hundred eggs and are very resistant towards inbreeding depressions [6], which means that even a small and isolated population can proliferate quickly.

With the decline of honeybee populations over the last decades [8, 9], ensuring crop pollination has become a large concern [10]. Besides bumble bees, mason bees are emerging beneficial arthropods that are used as managed pollinators in North America, Europe, and East Asia [11]. They belong to the genus *Osmia* (*Hymenoptera*: *Megachillidae*) with four different species that are used (*Osmia bicornis* & *Osmia cornuta* in Europe; *Osmia cornifrons* in Asia; *Osmia lignaria* in North America) [12–14].

Mason bees are solitary bees. They do not live in a social structure and do not have a queen or caste system. The above-mentioned species are univoltine and the adults are active in early spring [11]. During these months the females build nests within suitable cavities. These nests are stocked with a mix of pollen and nectar and if the provision is large enough the bee will lay an egg on top. When the cavity is filled and the nest is finished, it is closed with a mud plug. After 3–7 days the larvae emerge from the egg and start consuming the pollen. The bees spend the summer as pupae in a diapause. In late summer, metamorphosis is completed where the bees remain within their cocoons and enter diapause as adults. When spring temperatures are sufficiently high, bees will break their diapause and hatch. These adult mason bees have the potential to be an important pollinator of orchard crops, especially of early blooming plants like cherries and almonds [12, 15]. They fly at low temperatures [16], are specialized on fruit trees [11] and have a small flight radius [17].

This potential as orchard pollinators, resulted in a considerable amount of research that developed rearing, handling, and releasing methods for mason bees. Most of this work was done with *O. lignaria* in the USA [7, 18] but also for *O. cornuta* [19, 20] and *O. bicornis* [20] as well as *O. cornifrons* in Japan [21]. Rearing of these bees must be done outdoors, since it is impossible to provide enough flying space and floral resources in a greenhouse or a similar set-up. To our knowledge there is only one study that tried to establish an indoor rearing for *Osmia* bees [22]. In most of these systems the bees are entering diapause under controlled conditions under human supervision, prior to introduction into a suitable habitat (such as orchards shortly before the onset of bloom). Often the diapause of the bees have to be terminated beforehand, by exposing the bees to higher temperatures. The bees are provided with suitable nesting habitats and collect pollen from nearby sources and start to reproduce. The adults die after a few weeks and the larvae hatch inside the nesting material and start their development. These larvae stay outside during spring and summer and are normally collected in autumn. After finishing their development into adults, they are removed from their nesting material and are cleaned whilst still in their cocoon. The

cocoons containing the adult bees are then transferred to controlled temperatures for diapause. This process is described for *O. lignaria* in [7] with great detail, the handling of other *Osmia* species follows a similar manner. During this entire process the bees are constantly exposed to natural enemies. This increases to encounter of natural enemies drastically, as it is nearly impossible to prevent their entry into the breeding stock of mason bees, therefore considerable work was done in understanding the pest complex of *Osmia* bees [23]. Some of these natural enemies could cause significant damage but are easily detected and removed with proper cocoon care [7, 23]. However, since these methods were developed for small or medium-scale set-ups they can be challenging to implement on a large-scale. The effectiveness of such techniques is further diminished by the decentralized character of commercial mason bee rearings, often involving farmers and garden owners. Moreover, preventative care is not possible for parasitic wasps that attack mason bees. The most destructive parasitoids in commercial mason bees rearings are *Melittobia acasta* (*Hymenoptera*: *Eulophidae*) and several species of *Monodontomerus* (*Hymenoptera*: *Torymidae*) [7, 23]. These wasps lay their eggs inside the cocoons of the bees, the larvae consume the bee and overwinter inside the cocoons as larvae [23, 24]. These parasitoids provide a significant challenge due to their concealed lifestyle, where the parasitoid population could increase undetected. Therefore, it is essential to monitor the abundance of parasitoids constantly. Monitoring manually by opening the cocoons is not feasible for a large number of bees and can potentially endanger the bee inside the cocoon.

A non-invasive method for parasitoid detection is the use of x-ray, which is a non-invasive imaging method that allows for examination of different aspects of mason bee biology [7, 25–28] (Fig 2D). However, the use of x-ray radiography limits the number of cocoons to be evaluated, as only one layer of cocoons which should be orientated on their longitudinal axis allows for evaluation of the parasitoids. X-ray radiography provides a two-dimensional image of a three-dimensional situation and the standardization of the cocoon orientation, while imaging large numbers, is impossible. Accordingly, conventional x-ray radiography is inadequate for assessment of a large-scale mason bee rearing. Alternatively, computed tomography (CT), a volumetric x-ray-based technique, could be used. CT covers a three-dimensional volume and can therefore simultaneously scan multiple layers of cocoons. Accordingly, this results in a high throughput and the possibility for large-scale evaluations. CT based data allows to re-orientate the images in 3D, independent of the original object orientation. This allows questionable cocoons to undergo a detailed evaluation and reduces error rate compared to x-ray radiography. Even if large data assessment is possible based on CT, the limited availability of accessible CT scanners and the need for manual counting still limits large-scale use in mason bee rearings.

Traditionally, image processing is performed by a predetermined set of rules, based on available data, which are applied to novel data. Inspired by a self-organizing artificial network [29], LeCun et al. introduced the concept of convolutional neural networks as we know it today [30]. These models can, regardless of shift, recognize complex patterns. In the last decade, hardware improvements have allowed increased model complexity, supported by the availability of large datasets. Given the strong performance in many domains [31], this type of image analysis is applied to the challenge presented in this study.

In this study, we present a CT based approach for parasitoid detection in large-scale production of mason bees as a proof of concept. To our knowledge, this is the first time an automated screening method for a high number of cocoons is presented. Additionally, manual, and automated counting of parasitoid rates are described and compared, providing an incentive to up-scale future screenings and applications.

## Materials and methods

The European mason bee species *Osmia cornuta* (*Hymenoptera*: *Megachillidae*) was used as a model organism. Imaging was conducted with bees that developed in 2020 and were reared around Lake Zurich. The bees were brought to the rearing sites in mid-March and were allowed to nest. With the adults dying, flight activity of the bees stopped in late April, while their offspring developed inside the nesting substrate (*Arundo donax*, Giant reed stalks). In late August the bees had mostly finished their development into adults and the nesting material was collected and the cocoons were removed from it. In February 2021, the bees were imaged. The bees were taken from populations in diapause, which were kept at 0˚ prior to scanning with an in-between removal from the fridge and the scan was between 1 and 2 hours. Therefore, the bees were still in diapause during the scan, reducing the movement of bees inside the cocoons.

First trails for imaging parasitic wasps inside the cocoon of mason bees included conventional x-ray radiography. All x-rays were generated with the digital x-ray system (Fujifilm CR, Profect / Capsula CS). Computer tomography (CT) was performed using a Siemens Brilliance 16 slice CT scanner (slice thickness 0.8mm, KVP 120, mAs 151, spacing between slices 0.4mm) with a bone and soft tissue reconstruction protocol. The digital picture archiving system (PACS—Intellispace Philips) was used for DICOM image storage.

The randomly selected mason bee batch samples were scanned as a multiple layer volume in a custom-made box containing 12 compartments, each holding a maximum of around 600 cocoons of *Osmia cornuta* (Fig 1A). The cocoons were scanned in single layers marked with metal numbers, which facilitated discriminating between samples for manual counting (Fig 1B). Due to the low storage and scanning temperature movement was reduced to a minimum and all cocoon contents were manually identifiable without movement artifacts.

### Rapid and manual volumetric counting

All DICOM images were assessed with the freeware Synedra View Personal (Version 19.0.0.2; Synedra Information Technologies) on a Windows computer system or Horos (Horos Project, 64-bit medical image viewer for OS X, GNU Lesser General Public License, Version 3 (LGPL-3.0) on a Macintosh computer system. A multiplanar reformation (MPR) of each dataset was performed and allowed re-orientation and image alignment in all dimensions. Assessment of healthy and parasitoid cocoons was performed manually by two different approaches.

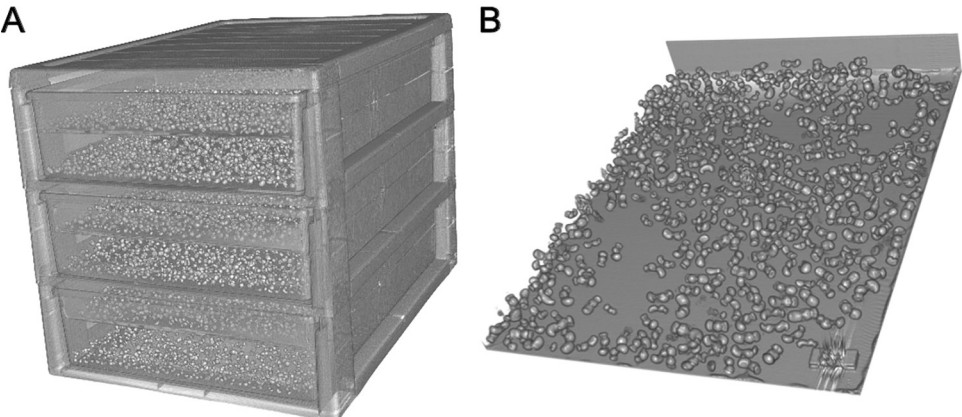

**Fig 1. 3D volume visualizations prior to AC.** A: 3D box filled with cocoons, C: Visualization of an individual sample.

After exporting suitable images into ImageJ [32], an experienced observer discriminated the sub populations by manual counting of one or several representative slices, which is coined manual rapid counting (RC). Manual rapid counting is considered the standard in industry. Another experienced observer performed a true manual counting in all 3D samples (which is considered the ground truth), this counting is coined volumetric counting (VC). All cocoons present in the volume were individually identified using 3D Slicer [33] in the VC method. In both the industry and individual counting methods, processing times were recorded (without image preparation) to allow for comparison with the automated counting method (AC).

## Automated counting

Given the state-of-the-art segmentation performance of UNet in biomedical image segmentation [34] it was used in this study. UNet contains an encoder and decoder path with skip connections to preserve low-level spatial features [34]. In this study, the automated counting (AC) segmentation algorithm consists of a 3D UNet architecture with residual units [35], developed using the MONAI framework. This framework is developed for deep learning in healthcare imaging and is freely available and open source [36]. The algorithm uses 3D volumes and multi-label maps as input. Training was performed on a local system, running a NVIDIA Tesla T4 GPU.

The algorithm had 5 layers with 16, 32, 64, 128 and 256 filters respectively, where each depth has two convolution layers followed by batch normalization and parametric rectifying linear activation. In all dimensions a stride of 2 was maintained. During training, 96x96x96 voxel input patches were clipped between -1000 and 600 Hounsfield units, followed by random rotating, flipping, and zooming along all axes. Moreover, random gaussian noise and smoothing were applied, followed by training with a learning rate of 1e-4 and weight decay of 1e-4 with a Dice loss and Adam optimizer. Cavities larger than 5 voxels in the output segmentation were filled following the thresholding of the output probability maps at threshold 0.8.

In training of the UNet, 19 volumes (manually annotated and counted in the same manner as the VC method) were used. Data was stratified to 15 volumes in the training set, and 4 in the validation set. Here, total cocoon content ranged between 148 and 606 in the volumes present in the training and validation set. Ultimately, performance was evaluated by comparing the counts of the AC (and the RC) with the counts according to the VC method on an independent test set of 12 samples.

## Micro-CT

On a few occasions, micro-CT has been used to study different bee species [37–42]. These studies focused on looking into anatomical details in individual specimens but were not aimed at large-scale studies on bee rearings. Hence, additional micro-CT scans were conducted on manually selected individual cocoons to visualize differences in parasitoid and healthy cocoons in detail. For micro-CT, an EasyTom XL Ultra 230–160 micro/nano-CT scanner was used (RxSolutions SAS, Chavanod, France). The scanner features a Hamamatsu open, reflection micro-focus x-ray tube and a Varian PaxScan 2520DX detector (flat panel with amorphous silicon and a CsI conversion screen; 1920 x 1536-pixel matrix; pixel pitch of 127 mm; 16 bits of dynamic range). As no predefined protocols exist on such micro-CT scanners for soft tissue bio-samples, expert judgment was used for defining the scan parameters. The tube was operated at 70 kV and a current of 150 mA. The voxel size of the CT scan was 8.3 μm for the parasitoid and 9.1 μm for the healthy cocoon. The images were acquired at 5.0 frames per second, averaging 2 frames per projection. The healthy bees in the cocoons and especially the larvae in the parasitoid cocoons tended, even if they were stored at very low temperatures prior to

scanning, to move during scanning. This, at the high resolution of the micro-CT, potentially results in motion artifacts. To minimize the chance of motion during the scan, the scan time was reduced to a minimum whilst still maintaining acceptable image quality.

## Macro photography of specimen

A focal length of 50mm was chosen to obtain a close photographic representation of the visually perceived specimen. Two studio flashes were set indirectly to the specimen resulting in emitting soft light, avoiding harsh shadows, and distracting highlights. Five to fifteen single images with individually set focus points were manually merged in post-production with Adobe Photoshop CC 2021, to enable a greater depth of field, enhancing the visibility of the specimen.

## Statistics

Statistical evaluation of the data was performed with the Python library Scipy [43]. Here, the Shapiro-Wilk test was used to investigate normality, followed by significance testing in the healthy cocoons by an independent sample t-test. Whereas the parasitoid cocoons and the parasitoid rate were tested by the Wilcoxon rank sum test for 2 independent measurements.

## Ethics statement

No vertebrate animal species were included in this study, and therefore no ethics statement is provided.

## Results

Conventional x-ray radiography and CT allowed to non-invasively distinguish between healthy bees and parasitoids inside the cocoons. X-ray radiography is applicable for individual cocoons or cocoons still inside their nesting substrate. Here, adult bees and parasitoids can be distinguished from each other (Fig 2D), whereas CT allowed for volume rendering and multi-

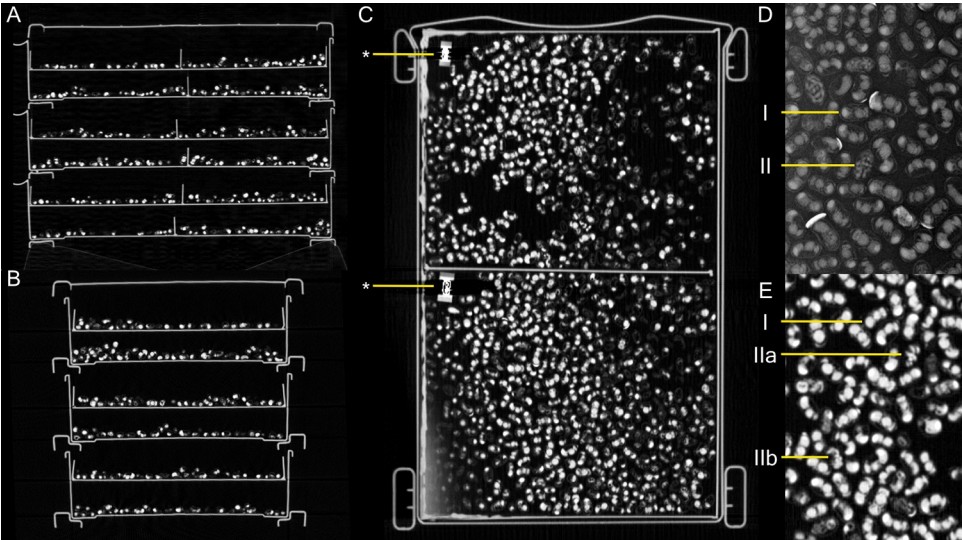

**Fig 2. CT scan of a customized plastic box containing around 6000 mason bees in 12 layers with metal numbers for identification of batches.** A-C: Three-dimensional view (latero-lateral, anterior-posterior, dorso-ventral), where the metal numbering is indicated with an *, D: Cocoon x-ray acquisition with healthy (I) and parasitoid (II) cocoons, E: Healthy (I) and two parasitoids which are easy (IIa) and challenging to detect (IIb), surrounded by adult bees in dorso-ventral view of CT acquisition.

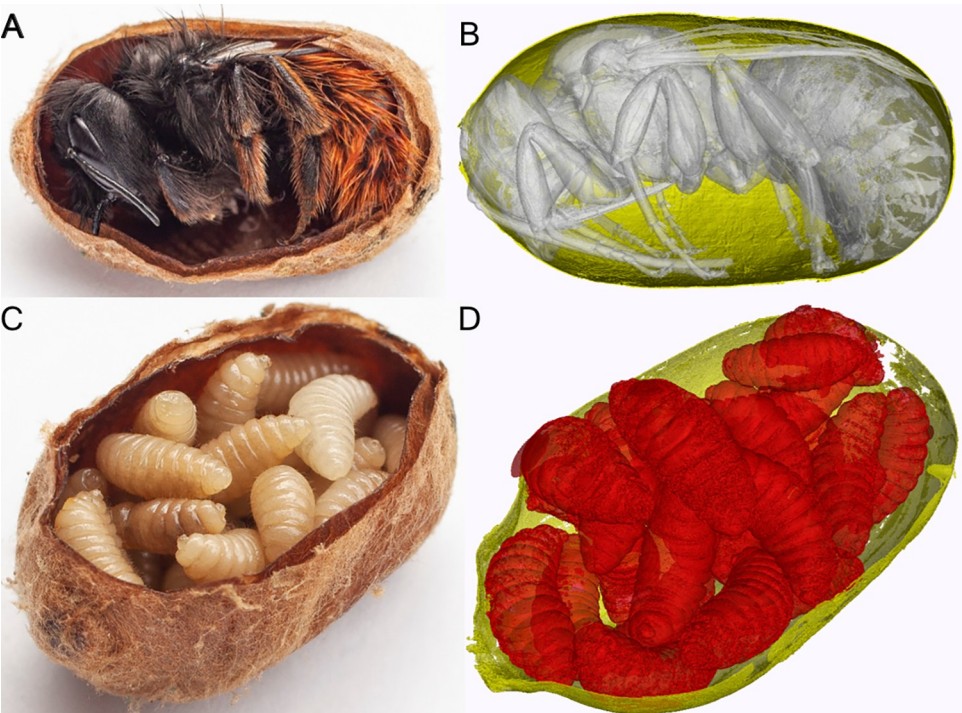

**Fig 3. Micro-CT cocoon comparison.** Healthy (A) and parasitoid (C) cocoons with 3D renderings and virtual incision based on high-resolution micro-CT results (B and D). The high resolution of the micro-CT can enable non-invasive examination of the internal anatomy of a healthy bee in its cocoon. The larvae in the parasitoid cocoon can also be clearly resolved in detail. Note that the larvae moved between the micro-CT and the preparation for photography inside the cocoon.

planar reconstruction (Fig 2A–2C & 2E). Micro-CT renderings of parasitoids and a healthy bee are presented in Fig 3 together with macro photographs. These images clearly illustrate the anatomical differences between a healthy and parasitoid cocoon that are further analyzed by radiographic imaging.

In 12 samples, a total of 5864 *Osmia cornuta* were manually counted within this study, according to the VC. Which is summarized in Table 1, together with the RC of 3371 and the AC of 5855. All healthy cocoon countings were normally distributed in each separate volume, only the manual RC was found to differ significantly ($p < 0.001$). 65 parasitoids were found by VC, compared to respectively 21 and 56 in the RC and AC. Again, the difference between the VC and the RC was found to be significant ($p = 0.004$). The aforementioned counts lead to parasitoid rates of 0.62%, 1.11% and 0.95% for the RC, VC and AC, respectively. Again, the RC and VC differed significantly ($p = 0.039$). Counting times are reported as 4.5 and 90 minutes per sample for the RC and VC, compared to 10 seconds for the AC. However, image preparation time was not included in the RC processing time and is therefore highly underestimated.

## Discussion

The AC presented in this study outperforms the traditional method of RC in classifying a large number of bee cocoons. Not only in accuracy the AC outperforms the RC method, but it is also roughly 25 times faster. Furthermore, it is ~500 times faster than identifying every cocoon individually, when compared to the VC. The presented method should be considered as a proof of concept, which can be developed in an effective, scalable, and economic method to

**Table 1. 12 samples of *Osmia cornuta* by *Monodontomerus spp.*, in bees recovered around lake Zurich (2020).**
Countings of identified healthy and parasitoid cocoons for the manual rapid (RC), manual volumetric (VC) and automated (AC) method. The ground truth, VC, is indicated in bold, * indicates significance.

| Sample number | | Healthy cocoons | | | Parasitoid cocoons | | | Parasitoid rate (%) | | |
|---|---|---|---|---|---|---|---|---|---|---|
| | | RC | VC | AC | RC | VC | AC | RC | VC | AC |
| | 1 | 221 | **301** | 291 | 0 | **2** | 4 | 0.0 | **0.66** | 1.37 |
| | 2 | 276 | **544** | 527 | 1 | **2** | 3 | 0.36 | **0.37** | 0.57 |
| | 3 | 362 | **690** | 664 | 0 | **3** | 2 | 0.0 | **0.43** | 0.3 |
| | 4 | 278 | **466** | 489 | 1 | **2** | 4 | 0.36 | **0.43** | 0.81 |
| | 5 | 268 | **378** | 375 | 2 | **6** | 6 | 0.75 | **1.59** | 1.6 |
| | 6 | 301 | **591** | 604 | 1 | **2** | 1 | 0.33 | **0.34** | 0.17 |
| | 7 | 279 | **466** | 487 | 8 | **18** | 15 | 2.87 | **3.86** | 3.1 |
| | 8 | 260 | **553** | 505 | 3 | **8** | 9 | 1.15 | **1.45** | 1.78 |
| | 9 | 334 | **402** | 399 | 0 | **7** | 3 | 0.0 | **1.74** | 0.76 |
| | 10 | 292 | **520** | 544 | 0 | **3** | 2 | 0.0 | **0.58** | 0.37 |
| | 11 | 216 | **407** | 388 | 2 | **2** | 2 | 0.93 | **0.49** | 0.52 |
| | 12 | 284 | **546** | 582 | 3 | **10** | 5 | 1.06 | **1.83** | 0.87 |
| | Total | 3371* | **5864** | 5855 | 21* | **65** | 56 | 0.62* | **1.11** | 0.95 |

improve the rearing of solitary bees. Due to the technical complexity of the approach, it could be of interest for companies that supply pollinators for large regions and might have centralized rearing facilities.

To our knowledge, this is the first study that indicates CT as a modality of choice for up scaling the screening of parasitoids within large batches of mason bee rearings. In contrast, conventional x-ray radiography provides limited space on the plates as they are available in three dimensions: small 18x24 cm, medium 24x30 cm and large 35x42 cm, therefore limiting the number of cocoons per x-ray acquisition. Accordingly, numerous x-rays acquisitions would be needed to analyze a representative sub-population of the bee rearing. Where micro-CT allows scanning of small volumes with very high resolution for detailed analysis, commercially available clinical CT scanners provide a scan volume for an uncountable number of bees with a sufficiently accurate image quality to distinguish healthy from parasitoid cocoons.

In this study, it was challenging to get all cocoons into focus simultaneously with the traditionally used manual rapid counting, as single CT slices were chosen. Accordingly, for some samples multiple slices had to be chosen for counting, resulting in an uncertainty to either double count cocoons due to ambiguous positioning, or not at all. When comparing this RC to the VC, almost double the number of cocoons are identified in the VC. However, the estimated parasitoid rate gives an approximation of the parasitoid rate as the amount of found parasitoids is also roughly half of the VC. A major downside is that it would take an inexperienced observer more time and most likely result in less accurate counting. Our automated approach has shown a significant time improvement when compared to VC, whilst obtaining more accurate results than RC. Unfortunately, analysis of CT data makes identification of the parasitoid species challenging; experience suggests that the underlying parasitoids belong to the genus Monodontomerus, where the most common parasitoids of mason bees are found [23].

Given these promising results, future work should focus on detecting other quality parameters, such as the number of dead bees (prepupae, pupae and adults), (un)healthy larvae, or the presence of *Anthrax anthrax* (*Dipetera*: *Bombyliidae*) inside the cocoons. For example, the potential of segmenting smaller structures in CT volumes is successfully presented in radiological literature [44, 45]. These techniques could eliminate the need for a custom-made box and potentially allow for scanning of a collection of cocoons in any arbitrary structure, further

decreasing the time and effort spent in scan acquisition. Additionally, a low x-ray absorption material such as styropor could be used instead of the custom-made box, as this will reduce the amount of noise.

Moreover, additional imaging can be used to further differentiate contents of either healthy or parasitoid cocoons. For this purpose, a healthy and a parasitoid cocoon were scanned by micro-CT, followed by comparing its 3D segmentation to classical macro photography, after dissection of these cocoons. The high resolution provided by micro-CT allows to identify underlying parasitoid species or to visualize mason bee anatomy details as well as to measure their structures on a micrometer level. Furthermore, micro-CT as a non-invasive imaging method is compatible with the potential research need of intact structures and is tackling research questions needing higher resolution. A drawback of micro-CT imaging is the limited sample size as the resolution decreases when increasing sample size. Moreover, due to its long scan duration it is more prone to movement artifacts, when compared to clinical CT. This prevents the use of micro-CT for commercial large-scale batch analysis, but it remains a useful complementary tool.

Despite the low percentage in populations from different origins, the presence of detected parasitoids shows the potential risk they can cause. An inadequately managed mason bee rearing could suffer high loss by these parasitoids. Therefore, well-managed mason bee rearings should include a method to detect parasitoids [7]. Moreover, the monetary value of pollination services has been estimated at 22 billion €, in Europe [3], suggesting future potential for such detection methods. Without early detection methods, the infestation would only be detected when significant damage has been caused. Combined with declines in bee populations due to climate change [46], this could be disastrous for the proliferation of bee rearings. While it is impossible to screen all cocoons, sampling of a representative sub-population would allow for detection of an accumulation of bee numbers over several years and stimulate the development of measures to counter that. The optimal size for such a sub-population needs to be determined in future research.

The methodology presented in this study defines a first step towards further development and up-scaling. The described methodology could be applied to separate large batches of cocoons based on their parasitoid rate. Determining if a specific batch is suitable for customer sale or not, this increases quality assurance for commercial breeders. Additionally, mere parasitoid cocoon detection for quality control might not be enough for all potential users and a system for the removal of the affected cocoons could be a powerful tool. In a first step this could be achieved by combining automated detection and the manual removal of affected cocoons. Similar to industrial sorting machines (i.e., seeds or tomatoes), an approach like this could be optimized by developing a system that automatically classifies and then sorts cocoons based on their content. Here, the cocoons could be spread on a conveyer belt, on which they undergo the proposed image analysis and automated sorting.

It should be possible to adapt the method to solitary bees with similar lifestyles as our species, especially *Osmia lignaria* and *Osmia cornifrons*. Also, the alfalfa leaf-cutter bee *Megachile rotundata* should be a suitable candidate for the application of this method. Retraining our AC method with novel annotated data allows the algorithm to recognize these additional species.

This method could be quite useful for research applications. It is an ideal method to assess the health and abundance of cavity nesting bees, as it is nondestructive, and the bees do not need to be removed from their nesting material. Due to the nature of a CT scan, it is also capable to detect the bees and their parasitoids in rather bulky material (e.g., blocks of wood) where x-rays may struggle. There are a couple of studies, that deal with this [25–28]. Thereby, the population dynamics of bees (and other insects) with a hidden lifestyle could be investigated in greater detail, as well as their interaction with their natural enemies.

While our study did not detect large differences between samples, suggesting an even parasitoid load, there are several additional applications of these techniques. As parasitoid rates are often linked to factors such as climate [47], vegetation [48] or habitat fragmentation [49], studies could be designed to better understand the effect of its exposure to mason bees. This can also be applied to other fields of ecological research to better understand the population dynamics of parasitoids with a hidden lifestyle that is not easily identified by visual inspections.

## Conclusion

The results of this study show an improved non-invasive parasitoid screening inside cocoons of mason bees and could potentially be extrapolated to other insect rearings. X-ray based imaging methods such as x-ray, CT and micro-CT enable to non-invasively differentiate between healthy and parasitoid cocoons in mason bee rearings. Furthermore, CT allows an efficient and effective screening for parasitoids, especially in large-scale batches. The automated counting surpasses manual rapid counting and could possibly be further improved by increasing the variability of scanned cocoons. Parasitoid rate was low within the randomly chosen samples and did not show significant differences. Influence of mason bee species, environmental factors, climate, and geographical differences on these results has to be shown in future research.

## Acknowledgments

The authors do not have any acknowledgements to make.

## Author Contributions

**Conceptualization:** Bart R. Thomson, Steffen Hagenbucher, Robert Zboray, Henning Richter.

**Data curation:** Bart R. Thomson, Steffen Hagenbucher, Robert Zboray, Michelle Aimée Oesch, Henning Richter.

**Formal analysis:** Bart R. Thomson, Robert Zboray.

**Investigation:** Bart R. Thomson, Robert Zboray, Michelle Aimée Oesch, Henning Richter.

**Methodology:** Bart R. Thomson, Robert Aellen, Henning Richter.

**Project administration:** Bart R. Thomson, Henning Richter.

**Resources:** Bart R. Thomson, Henning Richter.

**Software:** Bart R. Thomson.

**Supervision:** Henning Richter.

**Validation:** Bart R. Thomson.

**Visualization:** Bart R. Thomson, Michelle Aimée Oesch.

**Writing – original draft:** Bart R. Thomson, Steffen Hagenbucher, Robert Zboray, Michelle Aimée Oesch, Robert Aellen, Henning Richter.

**Writing – review & editing:** Bart R. Thomson, Steffen Hagenbucher, Robert Zboray, Michelle Aimée Oesch, Robert Aellen, Henning Richter.

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
