## [Decision Letter · Decision Letter 0]

2 Aug 2022

PONE-D-22-15307Automated computed tomography based parasitoid detection in mason bee rearingsPLOS ONE

Dear Dr. Thomson,

Thank you for submitting your manuscript to PLOS ONE. After careful consideration, we feel that it has merit but does not fully meet PLOS ONE’s publication criteria as it currently stands. Therefore, we invite you to submit a revised version of the manuscript that addresses the points raised during the review process.

We look forward to receiving your revised manuscript.

Kind regards,

David M. Lehmann, Ph.D.

Academic Editor

PLOS ONE

Journal Requirements:

Additional Editor Comments:

Thank you for submitting your manuscript to PLOS ONE. Please accept my apology for how long it took to get reviewer comments. I had a difficult time finding quslified peer reviewers for this manuscript. Then, after locating suitable reviewers, I left for family vacation. For those reasons, it took longer than typical for me to process the manuscript. Fortunately, we now have comments from two peer reviewers. Please carefully read their comments and provide a revised draft of the manuscript (one with track changes and a second 'clean version'). Once recieved, the revised manuscript will be distributed to peer reviewers for additional consideration.

Best,

Dave

Reviewers' comments:

Reviewer's Responses to Questions

**Comments to the Author**

1. Is the manuscript technically sound, and do the data support the conclusions?

Reviewer #1: Yes

Reviewer #2: Partly

2. Has the statistical analysis been performed appropriately and rigorously? 

Reviewer #1: Yes

Reviewer #2: No

3. Have the authors made all data underlying the findings in their manuscript fully available?

Reviewer #1: Yes

Reviewer #2: Yes

4. Is the manuscript presented in an intelligible fashion and written in standard English?

Reviewer #1: Yes

Reviewer #2: Yes

5. Review Comments to the Author

Reviewer #1: I appreciate the technical step forward with this paper.

A few thoughts:

Mono is one pest that we find.

The other is ascopharus (chalkbrood) that impacts the growing larvae mostly outside the silk spinning, but many times post-cocoon spinning.

We also just find dead, hardened larvae within cocoons.

In manual method, we place roughly 3-5,000wet cocoons on a flat LED light (24"x24") and visually scan cocoons that are opaque (larva dies before finished spinning), or have "not normal larva shape".

This research should suggest two further items to study:

1. software means to identify healthy larva and not healthy larva

2. mechanical means to remove unhealthy larva from reviewed larva

Ultimately cocoons would be spread out on a conveyor belt, analyzed, and as they cascade over the end of the belt, micro blasts of air would push the suspected not healthy cocoon out of the mix.

Reviewer #2: This manuscript addressed the use of computer-based tomography and AI to detect parasites and viable Osmia cells. While I believe this is interesting technology, due to the newness of this methodology, it would be important to have a greater background on a general overview of this process. Additionally, I think it is critical to talk about the validation of this model which I don’t this was adequately addressed. I also have so clarification comment throughout the manuscript.

Abstract:

You talk about insect husbandry, but I think this technology is more important for the agriculture management side then just general insect husbandry, I would like to see this discussed more in the manuscript.

Introduction:

You state “recent years” but really the use of bumble bees for pollination has been used for more than 100 years now and Osmia at least for the last 20.

At the beginning of the 4th paragraph remove “so called” mason bees are solitary bees there is no reason to have “so called” in there

At the beginning of the 5th paragraph, it there a citation for the first sentence

There are other papers that have looked at dealing with small scale production/ rearing of mason bees and scaling up, I think your paper would benefit from looking at these papers and discussing it here. Look for papers by Pitts-Singer, Boch, & Kemp.

Can you elaborate more on what we know about x-ray radiography and tomography for the use of bee or even other insects. Has this been used successfully in any species? Are there any studies that have used this technique on a small scale?

Methods:

What I felt was lacking in the methods was a test of accuracy. Do you know how many parasites you were starting with before they were placed in each board? What is the average percentages of parasites in this are in general?

How are you defining an “experienced observer”?

From what you wrote, it seems as though none of the cocoons were ever validated by actually opening the cocoons to see what is in them correct? Based on the images provided it looks like to would be hard to visually validate with out also extracting at least a subset of the cocoons for validations.

How would the manual analysis scale up to larger quantities.

Could the bees be scanned during diapause to limit the movement? How did the movement effect your overall success in detecting parasites?

With PLOS having such a general audience it seems like the AI section methods needs to be greatly expanded on.

Results:

Can you expand on why you chose to evaluate with the methods you choose, it seems like the RC method is not a very reliable method to use? How did these methods compare to actually hand counting?

p-values would include F statistic and degrees of freedom

What was the actual rate of how many parasites were in the population?

Discussion:

I’m not convinced the automotive method was more accurate.

Can you elaborate on this use for pollination, how do you see this expanding the mason bee industry? How easy would this be to use on other species such as O. lignaria, O. corniforms, ect ..? what about other species of interest such as the alfalfa leafcutting bee or Nomia

Can you discuss why you choose these two counting methods and how they are used in current practices

What would be the research applications of this, would they be limited too?

6. PLOS authors have the option to publish the peer review history of their article (what does this mean?). If published, this will include your full peer review and any attached files.

Reviewer #1: **Yes: **Dave Hunter, Crown Bees

Reviewer #2: No

---

## [Author Response · Author response to Decision Letter 0]

29 Aug 2022

We very much appreciate the feedback and constructive comments from the reviewer. We have prepared careful revisions with regard to the reviewers’ concerns and have highlighted the respective changes in the manuscript. Please find a detailed point-by-point response below.

General editorial comments:

We note that you have indicated that data from this study are available upon request. PLOS only allows data to be available upon request if there are legal or ethical restrictions on sharing data publicly. For more information on unacceptable data access restrictions, please see […]

Thank you for the heads up. We support open access to data and have updated the data availability statement on page 9 - as indicated in previous correspondence, we look forward to your editing.

Reviewer #1: 

I appreciate the technical step forward with this paper. A few thoughts:

Mono is one pest that we find. The other is ascopharus (chalkbrood) that impacts the growing larvae mostly outside the silk spinning, but many times post-cocoon spinning. We also just find dead, hardened larvae within cocoons. In manual method, we place roughly 3-5,000 wet cocoons on a flat LED light (24"x24") and visually scan cocoons that are opaque (larva dies before finished spinning), or have "not normal larva shape".

This research should suggest two further items to study:

1. software means to identify healthy larva and not healthy larva

2. mechanical means to remove unhealthy larva from reviewed larva

Ultimately cocoons would be spread out on a conveyor belt, analyzed, and as they cascade over the end of the belt, micro blasts of air would push the suspected not healthy cocoon out of the mix.

We thank the reviewer for the positive evaluation of our manuscript and appreciate the thoughtful comments. Based on the suggestions we have added an additional sentence in the fourth and second to last paragraph of the discussion:

Page 8: Here, the cocoons could be spread on a conveyer belt, on which they undergo the proposed image analysis and automated sorting.

Reviewer #2: 

This manuscript addressed the use of computer-based tomography and AI to detect parasites and viable Osmia cells. While I believe this is interesting technology, due to the newness of this methodology, it would be important to have a greater background on a general overview of this process. Additionally, I think it is critical to talk about the validation of this model which I don’t this was adequately addressed. I also have so clarification comment throughout the manuscript.

We thank the reviewer for the evaluation and have added an additional paragraph, on the background of the technology, in the introduction.

Page 3: Traditionally, image processing is performed by a predetermined set of rules, based on available data, which are applied to novel data. Inspired by a self-organizing artificial network [24], LeCun et al. introduced the concept of convolutional neural networks as we know it today [25]. These models can, regardless of shift, recognize complex patterns. In the last decade, hardware improvements have allowed increased model complexity, supported by the availability of large datasets. Given the strong performance in many domains [26], this type of image analysis is applied to the challenge presented in this study.

24. Fukushima K. Neocognitron: A self-organizing neural network model for a mechanism of pattern recognition unaffected by shift in position. Biological Cybernetics. 1980. pp. 193–202. doi:10.1007/bf00344251

25. LeCun Y, Bengio Y, Hinton G. Deep learning. Nature. 2015;521: 436–444.

26. LeCun Y, Boser B, Denker JS, Henderson D, Howard RE, Hubbard W, et al. Backpropagation Applied to Handwritten Zip Code Recognition. Neural Comput. 1989;1: 541–551.

Abstract:

You talk about insect husbandry, but I think this technology is more important for the agriculture management side then just general insect husbandry, I would like to see this discussed more in the manuscript.

Thank you for the comment, this would indeed be very valuable. We have added a short section in the introduction on page 2 about the application of insects in agriculture.

Introduction:

You state “recent years” but really the use of bumble bees for pollination has been used for more than 100 years now and Osmia at least for the last 20.

We have replaced “recent years” with “In the last decades”

At the beginning of the 4th paragraph remove “so called” mason bees are solitary bees there is no reason to have “so called” in there

We have removed the term “so called”.

At the beginning of the 5th paragraph, it there a citation for the first sentence

Thank you for the suggestions, we have changed it accordingly. 

There are other papers that have looked at dealing with small scale production/ rearing of mason bees and scaling up, I think your paper would benefit from looking at these papers and discussing it here. Look for papers by Pitts-Singer, Boch, & Kemp.

Thank you for the addition. We have added a few citations of relevant studies and expanded a bit on the rearing methods but decided to not go into great details of the rearing methods to keep the focus of the publication.

Can you elaborate more on what we know about x-ray radiography and tomography for the use of bee or even other insects. Has this been used successfully in any species? Are there any studies that have used this technique on a small scale?

We adapted the micro-CT paragraph accordingly to present previous work done on imaging of insects:

Page 5: On a few occasions, micro-CT has been used to study different bee species [32–37]. These studies focused on looking into anatomical details in individual specimens but were not aimed at large-scale studies on bee rearings.

Methods:

What I felt was lacking in the methods was a test of accuracy. Do you know how many parasites you were starting with before they were placed in each board? What is the average percentages of parasites in this are in general?

We agree with the reviewer that a test of accuracy is one of the most important aspects of such a paper. Since it was not fully clear, in the revised manuscript we have stressed that the volumetric counting (VC) method serves as a test of accuracy.

Page 5: Ultimately, performance was evaluated by comparing the counts of the AC (and the RC) with the counts according to the VC method on an independent test set of 12 samples.

How are you defining an “experienced observer”?

Thank you for the observations. These are observers who have prior experience in counting of cocoons. Due to the literal meaning of the expression, we have chosen to not further elaborate.

From what you wrote, it seems as though none of the cocoons were ever validated by actually opening the cocoons to see what is in them correct? Based on the images provided it looks like to would be hard to visually validate without also extracting at least a subset of the cocoons for validations.

We thank the reviewer for that note. Based on the acquired CT data, cocoon contents are well visible, and readouts are easy to differentiate. Prior experiments have validated x-ray imaging and cocoon content by the extracting. The suggestion of the reviewer is considered to be common knowledge based on published work (referred to below). Accordingly, it is not necessary to double check CT analysis by opening cocoons. Therefore, we consider the VC method as robust and suitable as ground truth.

Fischer, K., & Kornmilch, D. B. J. C. (2010). Einsatz von Mauerbienen zur Bestäubung von Obstkulturen.

How would the manual analysis scale up to larger quantities.

Thank you for thinking about the actual application of the concept that we’ve established. The current paper presents a proof of concept of an automated counting method that could be extrapolated in a manner that Reviewer #1 describes. The manual count (RC in our paper) is considered a comparison to the current industrial standard.

Could the bees be scanned during diapause to limit the movement? How did the movement effect your overall success in detecting parasites?

During the scan the bees were still in diapause. We added a sentence to clarify this. The bees were at 0° or 4°C before the scans and the time between removal from the fridge and the scan was between 1 and 2 hours. Given the low temperature that the cocoons are stored and scanned at, movement is reduced to a minimum. Moreover, no impeding movement artifacts were observed in the scanned cocoons. For clarity we have added the following sentence:

Page 4: Due to the low storage and scanning temperature movement was reduced to a minimum and all cocoon contents were manually identifiable without movement artifacts.

With PLOS having such a general audience it seems like the AI section methods needs to be greatly expanded on.

You are more than correct, we agree that the description was at a minimal level and have expanded accordingly on page 5.

Results:

Can you expand on why you chose to evaluate with the methods you choose, it seems like the RC method is not a very reliable method to use? How did these methods compare to actually hand counting?

We compare both the current industrial standard of counting (RC) and our novel method (AC) to the true number of present cocoons (VC). In the discussion on Page 7 we conclude that our method outperforms the current standard. The answer to your question will be found there.

p-values would include F statistic and degrees of freedom

What was the actual rate of how many parasites were in the population?

These results are presented in Table 1.

Discussion:

I’m not convinced the automotive method was more accurate.

Can you elaborate on this use for pollination, how do you see this expanding the mason bee industry? 

Thank you for the addition, we are discussing this in paragraph 7 of the discussion.

How easy would this be to use on other species such as O. lignaria, O. corniforms, ect ..? what about other species of interest such as the alfalfa leafcutting bee or Nomia

While we have not worked with the mentioned species, we assume that the knowledge is easily transferred to other Osmia species or Megachile rotundata. These species can all be handled in a similar way then O. bicornis or O. cornuta and the bees are enclosed and protected by tough cocoons and leaf material. Adapting the method to these species might need a few trials for recalibration, however this should be a relatively quick process. We added a sentence regarding this in paragraph 9 of the manuscript.

Nomia spp might be unsuitable for this method, due to it being a soil nesting species. However, we lack practical experience with Nomia and can therefore not judge it fairly.

Can you discuss why you choose these two counting methods and how they are used in current practices

We chose CT scanning because of its advantages compared to X-ray (limited number of cocoons per scan) or manual opening of cocoons (very time consuming and potentially harmful to the bees) - we mentioned this in the introduction. We are convinced that the new scan analysis protocol (AC) is both more accurate and faster in comparison to the standard procedure (RC). We have added a section (page 9) in the discussion to elaborate on the possible future advantages of the presented technique.

What would be the research applications of this, would they be limited too?

A novel 9th paragraph was added in the discussion to elaborate on the suggestion of the reviewer. Brevity was maintained to keep it within the scope of the current paper.

---

## [Decision Letter · Decision Letter 1]

21 Sep 2022

PONE-D-22-15307R1Automated computed tomography based parasitoid detection in mason bee rearingsPLOS ONE

Dear Dr. Thompson,

Thank you for submitting your manuscript to PLOS ONE. After careful consideration, we feel that it has merit but does not fully meet PLOS ONE’s publication criteria as it currently stands. Therefore, we invite you to submit a revised version of the manuscript that addresses the points raised during the review process.

We look forward to receiving your revised manuscript.

Kind regards,

David M. Lehmann, Ph.D.

Academic Editor

PLOS ONE

Journal Requirements:

Additional Editor Comments (if provided):

Reviewers' comments:

Reviewer's Responses to Questions

**Comments to the Author**

Reviewer #1: All comments have been addressed

Reviewer #2: All comments have been addressed. However, one more thing need to be addressed. Insects do not hibernate. They diapause; this needs to be corrected throughout the manuscript. 

2. Is the manuscript technically sound, and do the data support the conclusions?

Reviewer #1: Yes

Reviewer #2: Yes

3. Has the statistical analysis been performed appropriately and rigorously? 

Reviewer #1: Yes

Reviewer #2: N/A

4. Have the authors made all data underlying the findings in their manuscript fully available?

Reviewer #1: Yes

Reviewer #2: No

5. Is the manuscript presented in an intelligible fashion and written in standard English?

Reviewer #1: Yes

Reviewer #2: Yes

6. Review Comments to the Author

Reviewer #1: The Authors have met the various reviewers' comments and critique in an adequate manner. I look forward to progression of this technology.

Reviewer #2: I believe the authors have adequately addressed my comments. However, one more thing need to be addressed. Insect do not hibernate. They are in diapause; this needs to be corrected throughout the manuscript.

7. PLOS authors have the option to publish the peer review history of their article (what does this mean?). If published, this will include your full peer review and any attached files.

Reviewer #1: **Yes: **Dave Hunter

Reviewer #2: No

---

## [Author Response · Author response to Decision Letter 1]

22 Sep 2022

Dear Emily Chenette, Editor-in-Chief PLOS ONE,

Dear reviewers and editors of PLOS ONE,

We very much appreciate the feedback and time taken by the editor and reviewer. We have prepared careful revisions with regard to the reviewers’ concerns and have highlighted the respective changes in the manuscript. Please find a response below to the point raised by reviewer #2.

Reviewer #2: 

I believe the authors have adequately addressed my comments. However, one more thing need to be addressed. Insect do not hibernate. They are in diapause; this needs to be corrected throughout the manuscript.

#We would like to thank the reviewer for the additional time taken to handle our manuscript – we have made the corrections on page 3 and 4 of our manuscript.

We are very much looking forward to your reply,

Best regards on behalf of all authors,

Bart Thomson and Henning Richter

---

## [Editor Report · Decision Letter 2]

26 Sep 2022

Automated computed tomography based parasitoid detection in mason bee rearings

PONE-D-22-15307R2

Dear Dr. Richter,

We’re pleased to inform you that your manuscript has been judged scientifically suitable for publication and will be formally accepted for publication once it meets all outstanding technical requirements.

Kind regards,

David M. Lehmann, Ph.D.

Academic Editor

PLOS ONE

Additional Editor Comments (optional):

Thank you for quickly revising the manuscript to address the comment from Reviewer 2.
---

## [Editor Report · Acceptance letter]

3 Oct 2022

PONE-D-22-15307R2 

Automated computed tomography based parasitoid detection in mason bee rearings 

Dear Dr. Thomson:

I'm pleased to inform you that your manuscript has been deemed suitable for publication in PLOS ONE. Congratulations! Your manuscript is now with our production department. 

Kind regards, 

on behalf of

Dr. David M. Lehmann 

Academic Editor

PLOS ONE